# Comparing Repaired Subscapularis Tendon Integrity Using Ultrasound in Onlay Versus Inlay Reverse Shoulder Arthroplasty

**DOI:** 10.3390/jcm14020416

**Published:** 2025-01-10

**Authors:** Shri Kapilan, Marko Nabergoj, Alexandre Lädermann, Philippe Collin

**Affiliations:** 1CHP Saint Grégoire, 6 Boulevard de la Boutière, 35760 Saint-Grégoire, France; eqnox82@gmail.com (S.K.); docphcollin@gmail.com (P.C.); 2Institut Locomoteur de l’Ouest (ILO), 35760 Saint-Grégoire, France; 3Department of Orthopedics, Hospital Selayang, Batu Caves 68100, Malaysia; 4Valdoltra Orthopaedic Hospital, 6280 Ankaran, Slovenia; mmarkoj@gmail.com; 5Faculty of Medicine, University of Ljubljana, 1000 Ljubljana, Slovenia; 6Division of Orthopaedics and Trauma Surgery, Hôpital de La Tour, Rue J.-D. Maillard 3, 1217 Meyrin, Switzerland; 7Division of Orthopaedics and Trauma Surgery, Department of Surgery, Geneva University Hospitals, Rue Gabrielle-Perret-Gentil 4, 1211 Geneva, Switzerland; 8Faculty of Medicine, University of Geneva, Rue Michel-Servet 1, 1211 Geneva, Switzerland; 9FORE (Foundation for Research and Teaching in Orthopedics, Sports Medicine, Trauma, and Imaging in the Musculoskeletal System), Avenue J.-D. Maillard 3, 1217 Meyrin, Switzerland; 10American Hospital of Paris, 55 boulevard du Château, 92200 Neuilly-sur-Seine, France; 11Clinique Victor Hugo, 5 Bis Rue du Dôme, 75116 Paris, France

**Keywords:** prosthesis, outcomes, results, complications, design, healing

## Abstract

**Background:** The importance of the subscapularis tendon in reverse shoulder arthroplasty (RSA) has been increasingly emphasized lately. Recent studies have indicated that a repaired subscapularis tendon has better functional outcomes. This study is aimed at comparing the healing rate of repaired subscapularis tendons between onlay and inlay Bony Increased Offset-Reversed Shoulder Arthroplasty (BIO-RSA). **Methods:** This retrospective comparative review covers all patients who underwent BIO-RSA at a single center, comprising 189 cases performed by on a single surgeon from January 2012 till December 2021. We included all patients who underwent subscapularis tenotomy repair and who had a tendon ultrasound (US) examination at six months postoperatively (as requested in this single surgeon’s usual protocol). These patients were divided into two comparable groups, an onlay group and an inlay group. Healing status was determined using the Sugaya classification, with healed subscapularis tendons defined as having Sugaya type I–III integrity and the unhealed tendons as having Sugaya type IV and V integrity. **Results:** In total, 2 patients were excluded because ultrasound was not performed (they missed their appointment); 187 patients were evaluated; 98 patients underwent an onlay BIO-RSA; and 89 patients underwent an inlay BIO-RSA. The healing rate of the repaired subscapularis tendon was 73% in the onlay group and 56% in the inlay one (*p* = 0.020). **Conclusions:** The onlay systems may enhance subscapularis tendon healing compared to the inlay one, possibly due to the preserved intramedullary vascularity and the near-normal tendon excursion that can be achieved by the onlay system. Choosing an onlay design can minimize bone cuts during RSA, while achieving good subscapularis tendon healing.

## 1. Introduction

Two distinct humeral designs have been used extensively with reverse shoulder arthroplasty (RSA): the onlay design, in which the humeral tray sits on the metaphysis of the proximal humerus at the level of the humeral neck cut [1], and the inlay design, in which the humeral tray is seated within the metaphysis. Theoretically, inlay systems adopt inferior positioning and a lower neck–shaft angle, making the pivot point for the range of motion closer to the center of rotation in the humeral head, maximizing the impingement-free range of motion throughout the coracoacromial arc [2]. In contrast, onlay systems may be more bone-preserving and facilitate modularity and convertibility; however, such designs result in additional humeral component lateralization and lengthening [3,4].

There is still ongoing controversy regarding subscapularis tendon integrity after RSA. The recent literature shows that repairing the subscapularis tendon in RSA can improve active internal rotation [5], especially in restoring the strength and enhancing shoulder stability [6], but there are also reports showing that shoulder strength and function are not significantly impacted by this form of repair [7,8,9,10]. However, only a few studies have assessed the integrity of this tendon during postoperative follow-ups [5,11,12,13,14]. Moreover, it is currently unknown whether the prosthetic design influences subscapularis tendon healing after reverse shoulder arthroplasty.

The goal of this study was to determine which type of RSA implant offers better healing potential for repaired subscapularis tendons. In this study, we compared the healing rate of the subscapularis tendon in onlay and inlay Bone Increase Offset (BIO)-RSAs. We hypothesized that both designs would have similar subscapularis healing rates.

## 2. Materials and Methods

### 2.1. Study Design, Data Collection, and Ethical Committee Approval

We retrospectively identified all patients who underwent BIO-RSA performed by a single surgeon (P.C.) from January 2012 to December 2021. This was a comparative retrospective review between two comparable groups, an onlay group and an inlay group. Approval from the local institutional review board was obtained under the reference number CERC-VS-2020-11-1, and all patients provided written informed consent for their data and images to be used in this research and publication. The inclusion criteria were subscapularis repair during BIO-RSA for cuff tear arthropathy, massive cuff tear, and glenohumeral osteoarthritis. A subscapularis tendon was classified as non-repairable if grade 3 or 4 fatty infiltration [15] was present, and external rotation was limited to below 0 degrees. Patients were grouped by implant design (onlay vs. inlay). Exclusions included different surgical indications, psychiatric constraints hindering informed consent or literacy, clinical follow-up periods of less than two years, absence of a six-month post-surgery subscapularis tendon ultrasound (US), and incomplete documentation.

### 2.2. Surgical Procedure

BIO-RSA procedures were performed with patients in the beach chair position, using an interscalene block at the operated site along with general anesthesia. A standard deltopectoral approach was used. Tenodesis of the long head of the biceps was performed at the level of pectoralis major tendon insertion. In cases where the subscapularis was intact, subscapularis tenotomy was performed at the midsubstance, approximately one centimeter medially to the insertion site following circumflex scapular artery ligation. The humeral site was prepared first, with patients receiving a 145-degree neck–shaft angle stem (Flex Shoulder system; Stryker, Bloomington, MN, USA) for the onlay system (Figure 1) and a Grammont-style 155-degree neck–shaft angle RSA (Aequalis Reverse II; Stryker, Bloomington, MN, USA) for the inlay system (Figure 2).

Glenoid preparation was performed after humeral preparation. A glenosphere was implanted using the standard technique described by the manufacturer.

Before placing the humeral stem, two 2 mm drill holes were created in the bicipital groove spanning from superior to inferior aspects of the subscapularis insertion. Two additional holes were made in the metaphysis, just medially to the tendon stump. Two high-strength #2 loop sutures were passed laterally to medially, entering the bicipital groove, creating loops to hold the humeral stem, and exiting medially. The inferior two sutures exited through the medial metaphyseal holes, and the two superior sutures exited below the collar of the humeral stem. The two-loop suture limbs were then threaded through the medial subscapularis tendon. The tendon was reduced to its anatomical position, securing it with at least four simple #2 sutures tied with a 7-throw surgeon’s knot. Two racking hitches were created laterally on the loop sutures. One superior limb and one inferior limb were then passed through the superior racking hitch knot, locking the repair with two half-hitches. The same process was repeated for the inferior racking hitch using the remaining two suture limbs (Figure 3, Figure 4, Figure 5, Figure 6, Figure 7 and Figure 8) [16]. After implant insertion, shoulder stability and range of motion were assessed.

### 2.3. Postoperative Rehabilitation

Both groups followed the same postoperative protocol. A sling immobilized the shoulder in an internal rotation for four weeks to protect the repaired subscapularis tendon [17]. The sling could be removed for hygiene and light mobilization exercises, such as pendulum exercises. Hand and elbow use for simple daily activities was encouraged while avoiding external rotation. At four weeks postoperation, the sling was removed, and gradual shoulder activity resumption was guided by the patient’s tolerance.

### 2.4. Ultrasound US Evaluation

All patients included underwent a six-month follow-up US assessment conducted by a board-certified musculoskeletal radiologist specialized in postoperative rotator cuff evaluation. The assessments were conducted with the patients in a seated position [18]. The Sugaya classification, initially developed for the MRI-based evaluation of tendon integrity post-rotator cuff repair, was used with similar efficacy for US assessments [19,20]. In this analysis, patients were categorized as “healed” (Sugaya types I–III) or “not healed” (Sugaya types IV and V) [21].

### 2.5. Statistical Analysis

Statistical analysis was performed using R software (version 3.6.1; Foundation for Statistical Computing, Vienna, Austria). Descriptive statistics were computed for the variable of interest. A chi-square test was used to compare RSA design by gender, healing rate, and age group separately. The healing rate was also analyzed by age group. To explore the relationship between healing, design, and diagnosis, a logistic regression model included the outcome variable “healed” and the following explanatory variables: age, diagnosis, and RSA design.

## 3. Results

From the initial cohort of 189 patients, 2 were excluded due to missing US assessments (they missed their appointment). This left 187 patients available for clinical evaluation. Among these individuals, there were 72 males and 115 females, with a mean age of 71.3 years (range, 48 to 91 years). The mean follow-up period was two years (range, 2 to 5 years). The onlay group comprised 98 patients (37 males, 61 females, mean age 69.8 years range 48 to 86 years), while the inlay group included 89 patients (35 males, 54 females, mean age 72.9 years, range 56 to 91 years). The healing rate was 73% in the onlay group and 56% in the inlay group (*p* = 0.020) (Figure 9).

The demographic characteristics, diagnoses, and radiological results are presented in Table 1 and Table 2.

Subscapularis tendon healing rates were consistent across all age groups and unaffected by diagnosis (Table 3).

Table 4 shows the logistic regression model, with onlay designs and osteoarthritis yielding better tendon healing chances.

To provide more information on the precision of the estimates, we reported the confidence intervals (CIs) alongside the *p*-values in a binary logistic regression analysis (Table 5).

Among the significant factors, we find that the diagnosis of MCT (massive cuff tear) is associated with a reduced probability of healing compared to that OA. The MCT group has an approximately 62% lower chance of recovery than the OA group (OR = 0.376, *p* = 0.022).

It also appears that the design of the onlay implant is associated with an increased probability of healing compared with that of the inlay implant. The onlay implant design was associated with a higher likelihood of healing compared to the inlay design (OR = 2.322, *p* = 0.024).

As for non-significant factors, the age, sex and side of the body were not statistically significant in this model. And the diagnosis of CTA (cuff tear arthropathy) has no significant effect on healing compared to OA.

For clinical implications, the choice of onlay design seems to be a more favorable option for patient healing. Patients with a diagnosis of MCT (massive cuff tear) may require specific or alternative interventions to improve the chances of recovery.

## 4. Discussion

Our study aimed to identify the superior BIO-RSA implant for subscapularis tendon healing. Contrary to our hypothesis, our results indicated that the onlay design is recommendable over the inlay system. Many reasons could explain our findings.

Various factors can impact postoperative tendon healing, including patient age, muscle changes, repair type, tension during and after repair, and tendon vascularity [22,23]. Of these factors, we could not demonstrate that patient age significantly influences RSA tendon healing. Azar et al. found that patients over 55 of age have poorer rotator cuff tendon healing rates [24]. A similar outcome was also observed in a study performed by Boileau et al. in 2005 [25]. It is known that as we age, the number of tendon stem cells decreases, and the remaining cells tend to have altered phenotypes [26,27,28,29]. Aging is also known to decrease extracellular matrix protein turnover, which in turn causes altered hemostasis and angiogenesis in tendons eventually making it more difficult for them to heal [30,31]. Moreover, fatty infiltration, type of repair, and tension on the reconstruction were the same in the two groups, as we excluded significant fatty infiltration, used identical tendon repairs, and did not repair subscapularis under tension. Consequently, the observed differences in tendon healing rate should be explained by either vascularity or biomechanical changes related to prosthetic designs.

### 4.1. Vascularity

Vascularity has received little attention in clinical studies, yet it may play a significant role in tendon healing. The primary blood vessels that contribute to subscapularis vascularization are classically the suprascapular artery in 36% of cases, the subscapular artery in 96% of cases, the circumflex scapular artery in 9% of cases, the thoracodorsal artery in 39% of cases, and the posterior circumflex humeral artery in 4% of cases, whereas the lateral thoracic artery supplied it in 14% of cases [32]. According to Youn et al. [33], the myotendinous junction of subscapularis mainly receives their vascular supply from the subscapular artery, circumflex scapular artery and posterior circumflex humeral artery. The healing of the subscapularis muscle can be explained by the complex plexiform of anastomoses formed by vessels of the subscapular artery, lateral thoracic artery, perforating branches and multiple tributaries around the muscle. But we still cannot explain why the healing in onlay systems is better compared to that in inlay systems. Minimal bone cuts with no intramedullary reaming and less periosteal stripping may provide a better foundation for tendon healing and the revascularization of tendons [34]. However, this needs further exploration, especially with regard to the vasculature of transected tendon and its healing potential.

### 4.2. Design

Secondly, the neck–shaft angle of the onlay system’s 145° neck–shaft angle allows for near normal subscapularis tendon excursion during movement. This could be explained by the variability in the subscapularis tendon healing rate reported in other studies [5,11], with a better healing rate being observed in studies using humeral implants with varus neck–shaft angles [13,35]. Repairing the subscapularis tendon without tension does play an important role in healing as it reduces the stress of tendon healing and causes less microtears in the tendon, leading to better healing [36,37,38]. Low healing rates in the setting of RSA can be attributed to the distalization [38] of the subscapularis tendon as well as poor native biologic environments in the case of rotator cuff arthropathy [39].

### 4.3. Present Healing Rate Compared to the Literature

Our study reveals a 73.5% healing rate for the onlay design compared to only 56% in the inlay group. Erickson et al. reported an 83.3% healing rate using double-row subscapularis peel repair with a 135-degree humeral neck–shaft angle [13,35]. Collin et al. reported a subscapularis tendon healing rate of approximately 52.6% [5]. There is variability in the reporting of subscapularis healing [5,11,13,40], with higher rates associated with humeral implants featuring more varus neck–shaft angles [35].

The choice of repair method impacts subscapularis tendon healing after RSA. Lesser tuberosity osteotomy (LTO) is considered biomechanically superior with a good healing rate [41]. Louis-Philippe Baisi’s study on total shoulder replacement found a higher subscapularis tendon healing rate with tenotomy (98%) compared to peel repair (75%), highlighting the benefits of the tendon-to-tendon method [42]. The healing rate of the subscapularis tendon using different techniques is not similar in all the studies, but they do show good results and a good range of motion [43,44]. There are also other factors that could influence the healing of the subscapularis tendon, such as bone and tendon quality. Healing rates tend to improve with implants featuring varus neck–shaft angles for RSA, as demonstrated in prior research [45].

### 4.4. Strengths and Limitations

Our study has several strengths. We had a large cohort of nearly 100 patients, evenly distributed among groups, with consistent patient characteristics and diagnoses. All procedures were performed by a single surgeon using the same approach, tenotomy, and the same repair technique. Additionally, our six-month postoperative US assessments were blinded. However, we did not analyze other RSA system components or patient factors like co-morbidities, habits, and environmental influences on subscapularis healing.

## 5. Conclusions

Onlay systems may enhance subscapularis tendon healing compared to inlay ones, possibly due to the preserved intramedullary vascularity and the near-normal tendon excursion that can be achieved by onlay systems. Opting for onlay designs during RSA could minimize bone cuts and enhance subscapularis healing. Further research should explore how vascular and biomechanical factors interact to influence tendon healing.

## Figures and Tables

**Figure 1 jcm-14-00416-f001:**
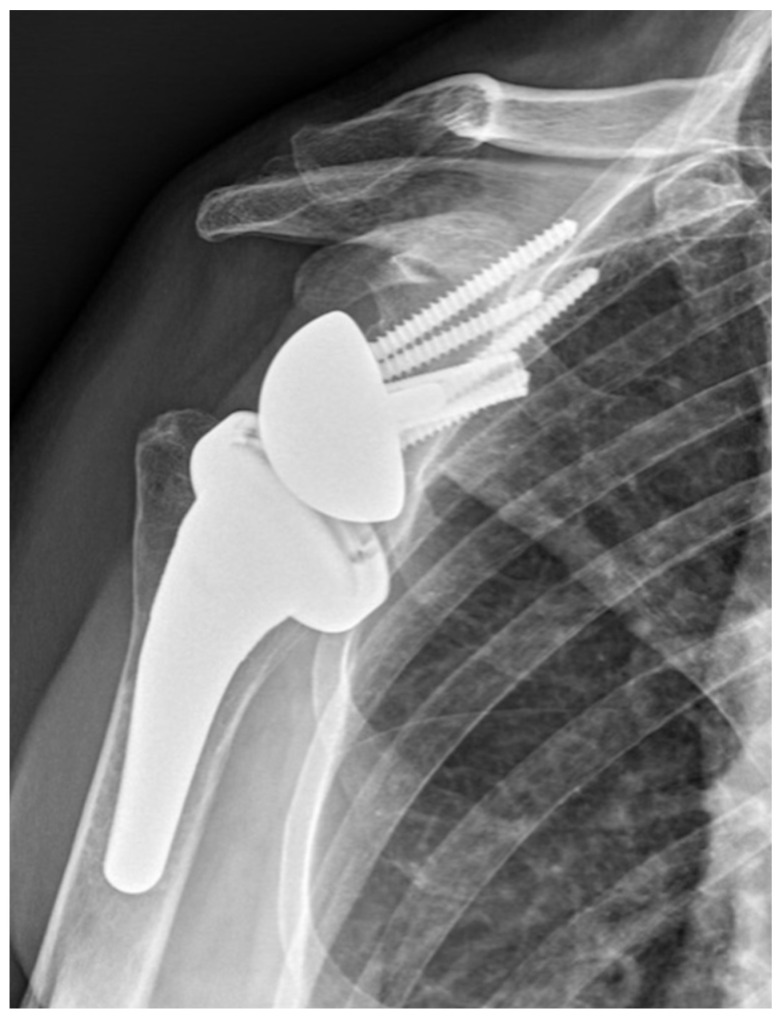
Onlay design.

**Figure 2 jcm-14-00416-f002:**
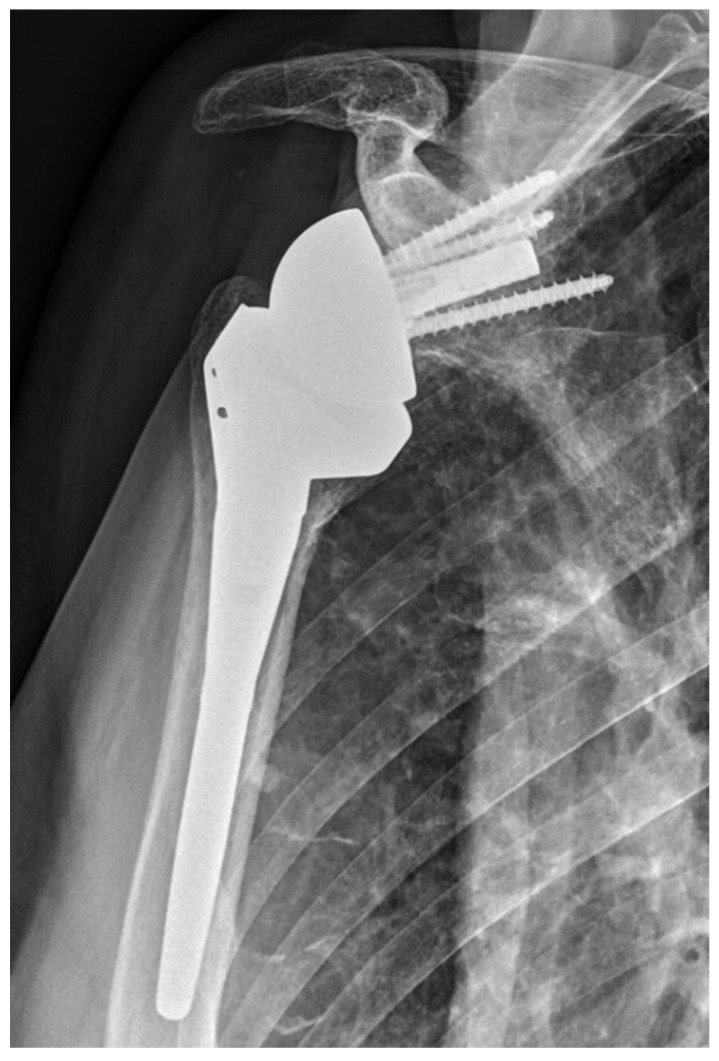
Inlay design.

**Figure 3 jcm-14-00416-f003:**
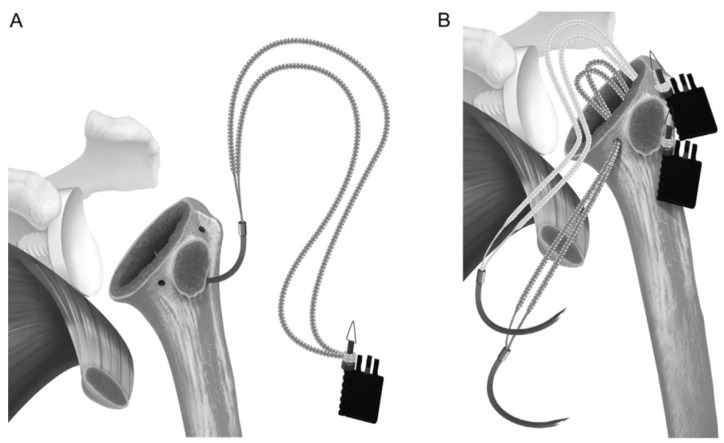
(**A**) A suture tape prefashioned with a half racking suture on the end is passed laterally to medially through the inferior 2 holes, and (**B**) a separate suture is passed through the superior hole. Reproduced from Denard et al. [16], with permission.

**Figure 4 jcm-14-00416-f004:**
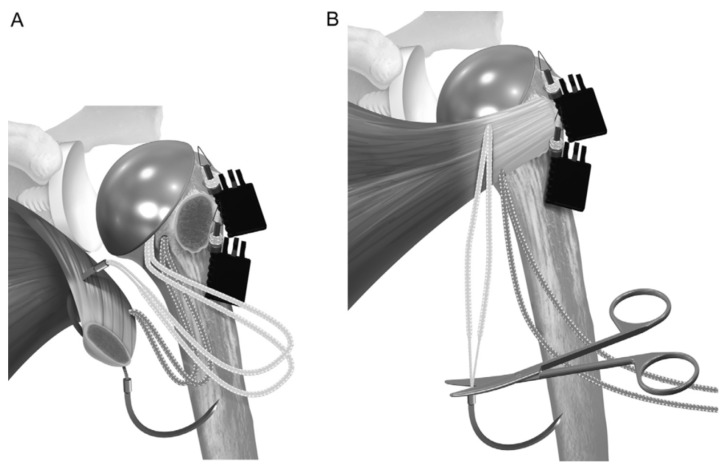
The stem is placed so that the sutures pass around the prosthesis. (**A**) The sutures are passed through the subscapularis tendon, and (**B**) the wedged ends are cut to provide access to 4 free limbs. Reproduced from Denard et al. [16], with permission.

**Figure 5 jcm-14-00416-f005:**
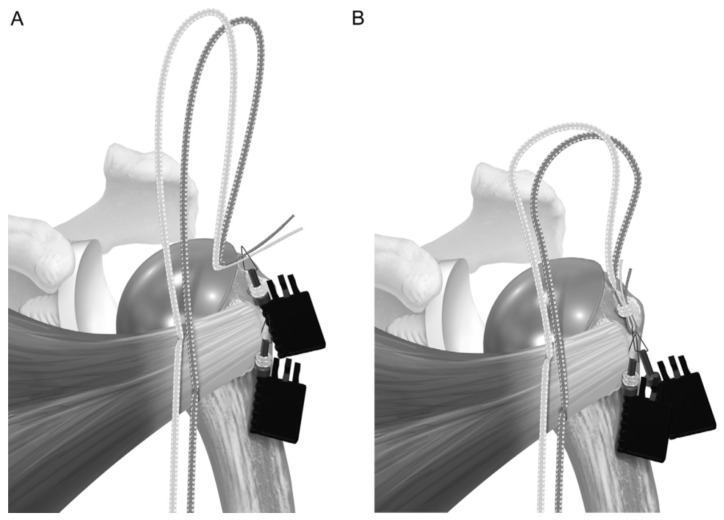
(**A**) One suture limb from each pair is selected and (**B**) passed through the prefashioned half racking suture. Reproduced from Denard et al. [16], with permission.

**Figure 6 jcm-14-00416-f006:**
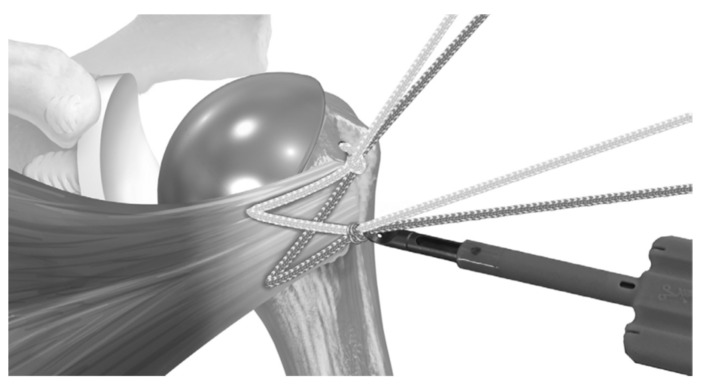
The suture limbs passed through the half racking suture are tensioned. Tensioning is conducted under visual inspection. Reproduced from Denard et al. [16], with permission.

**Figure 7 jcm-14-00416-f007:**
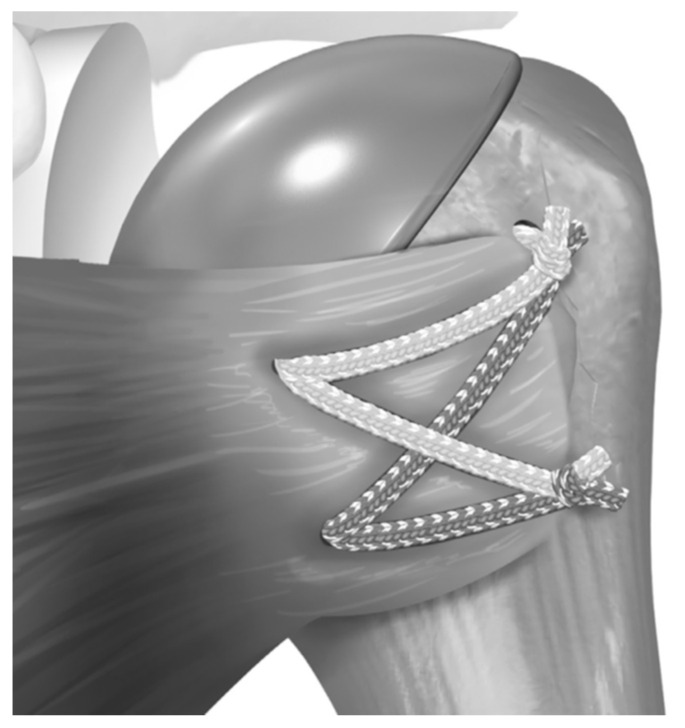
Final repair. Reproduced from Denard et al. [16], with permission.

**Figure 8 jcm-14-00416-f008:**
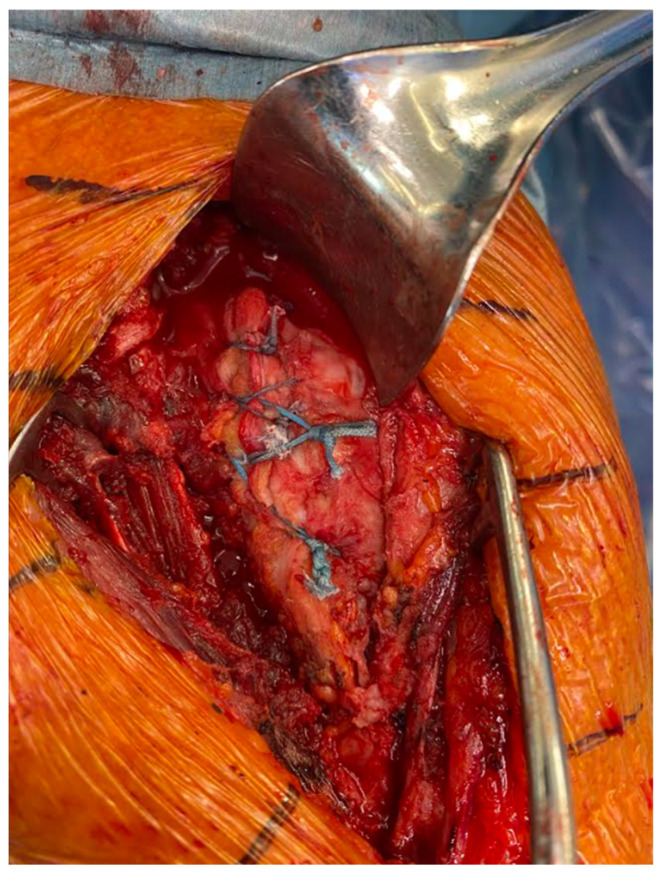
Transosseous of the subscapularis tendon in RSA repaired.

**Figure 9 jcm-14-00416-f009:**
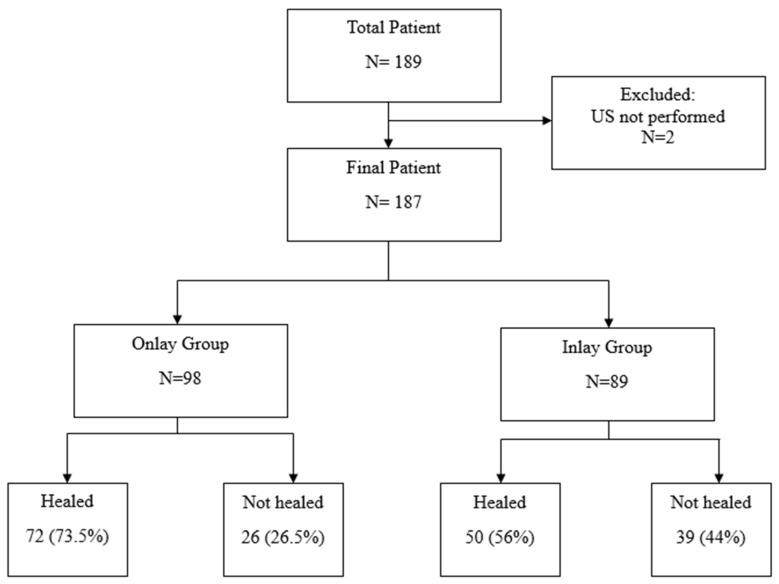
Flow chart.

**Table 1 jcm-14-00416-t001:** Demographic data.

Variable	Subcategory	Inlay RSA *n* (%)	Onlay RSA *n* (%)	Total *n* (%)	*p*-Value
Sex	Male	35 (39)	37 (38)	72 (38)	*p* = 0.82
	Female	54 (61)	61 (62)	115 (62)	
Age	<60	3 (17)	15 (83)	18 (10)	*p* = 0.04
	60–69	27 (48)	29 (52)	56 (30)	
	70–79	43 (51)	42 (49)	85 (45)	
	≥80	16 (57)	12 (43)	28 (15)	
Diagnosis	OA	24 (31)	54 (69)	78 (42)	*p* = 0.06
	CTA	53 (78)	15 (22)	68 (36)	
	MCT	12 (29)	29 (71)	41 (22)	

Legend: OA = osteoarthritis; CTA = cuff tear arthropathy; MCT = massive cuff tear; RSA = reverse shoulder arthroplasty.

**Table 2 jcm-14-00416-t002:** Radiographic results.

Criteria	Subcategory	Healed *n* (%)	Non Healed *n* (%)	Total *n* (%)	*p*-Value
RSA Design	Inlay	50 (56)	39 (44)	89 (100)	*p* = 0.013
	Onlay	72 (73)	26 (27)	98 (100)	
SSC Type	Type I	19 (15) *	0 (0)	19 (10) ***	
	Type II	91 (75) *	0 (0)	91 (49) ***	
	Type III	12 (10) *	0 (0)	12 (6) ***	
	Type IV	0 (0)	12 (18) **	12 (6) ***	
	Type V	0 (0)	53 (82) **	53 (29) ***	

Legend: RSA = reverse shoulder arthroplasty; SSC = Subscapularis. * Percentage of healed tendons. ** Percentage of non-healed tendons. *** Percentage of those healed using the RSA design.

**Table 3 jcm-14-00416-t003:** Healing of subscapularis tendons identified using US assessment according to age group and preoperative diagnosis.

Criteria	Subcategory	Healed *n* (%)	Non Healed *n* (%)	Total *n* (%)	*p*-Value
Age	<60	14 (78)	4 (22)	18 (100)	*p* = 0.190
	60–69	38 (68)	18 (32)	56 (100)	
	70–79	49 (58)	36 (42)	85 (100)	
	≥80	21 (75)	7 (25)	28 (100)	
Diagnosis	OA	58 (74)	20 (26)	78 (100)	*p* = 0.059
	CTA	42 (62)	26 (38)	68 (100)	
	MCT	22 (54)	19 (46)	41 (100)	

Legend: OA, osteoarthritis; CTA, cuff tear arthropathy; MCT, massive cuff tear.

**Table 4 jcm-14-00416-t004:** Logistic regression model with healing as the outcome variable.

	Category	OR	95% Confidence Interval	*p* Value
Age	60–69/<60	0.6	0.17–2.09	*p* = 0.425
	70–79/<60	0.39	0.12–1.28	*p* = 0.119
	≥80/<60	0.86	0.21–3.48	*p* = 0.833
Diagnosis	OA/MCT	2.5	1.12–5.6	*p* = 0.026
	CTA/MCT	1.39	0.6–3.23	*p* = 0.443
	Onlay/Inlay	2.16	1.20–3.87	*p* = 0.010

Legend: OA, osteoarthritis; CTA, cuff tear arthropathy; MCT, massive cuff tear.

**Table 5 jcm-14-00416-t005:** Results of binomial logistic regression: healing (model coefficients).

95% Confidence Interval							
Predictor	Estimate	Lower	Upper	SE	Z	*p*	Odds Ratio
Intercept	0.914	−0.500	2.329	0.722	1.267	0.205	2.495
Age:							
≥80–<60	−0.161	−1.660	1.338	0.765	−0.211	0.833	0.851
60–69–<60	−0.327	−1.620	0.967	0.660	−0.495	0.620	0.721
70–79–<60	−0.851	−2.099	0.396	0.637	−1.337	0.181	0.427
Diagnosis:							
MCT–OA	−0.977	−1.815	−0.140	0.427	−2.287	0.022	0.376
CTA–OA	−0.185	−0.986	0.617	0.409	−0.452	0.651	0.831
Design:							
1–0	0.842	0.110	1.574	0.373	2.256	0.024	2.322
Sex:							
F–M	0.359	−0.297	1.015	0.335	1.073	0.283	1.432
Side:							
1–0	−0.148	−0.809	0.514	0.338	−0.438	0.661	0.862

Note: Estimates represent the log odds of “healing = 1” vs. “Healing = 0”. Legend: SE = standard error; Z = Z value (or Z statistic, which measures the statistical significance of each predictor in the model); *p* = *p* value; OA = osteoarthritis; CTA = cuff tear arthropathy; MCT = massive cuff tear; F = female; M = male. Design: 1 (for onlay) and 0 for inlay. Side: 1 for right and 0 for left.

## Data Availability

All relevant data are included in the present manuscript. For additional information, please contact cdaniel.clinicalresearch@yahoo.com.

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
