# Peer review of "Comparing Repaired Subscapularis Tendon Integrity Using Ultrasound in Onlay Versus Inlay Reverse Shoulder Arthroplasty"

_jcm, 2025, doi:10.3390/jcm14020416_

Round 1

Reviewer 1 Report

Comments and Suggestions for Authors

The Authors aimed to compare the healing rate of subscapularis tendon between onlay and inlay Bony Increased Offset-Reversed Shoulder Arthroplasty (BIO-RSA). 

Although the topic is interesting, this retrospective study presents major biases.

Was ultrasound performed by the same radiologist and with the same device?Ultrasounds is extremely dependent by the operator and might have potentially influenced their findings. Thus, the rate of subscapolaris healing might be different. In particular, the Authors used an MRI based classification. I strongly believe that MRI is required.

Was the subscapolaris damaged before surgery in all cases?Or was it cut by the procedure itself?Also, the type of lesion might be different over the series.

Tables are very disorganized and confusing.

Please provide graphic representation of inlay and outlay designs.

Conclusions are not supported by results. These, are only hiopthesis of the Authors.

Comments on the Quality of English Language

Many grammar and syntax errors. Papers disorganized and confusing.

Author Response

Reviewer #1:

Thank you for your comments. Please see our answers below.

Comment 1: Was ultrasound performed by the same radiologist and with the same device? Ultrasounds is extremely dependent by the operator and might have potentially influenced their findings. Thus, the rate of subscapolaris healing might be different. In particular, the Authors used an MRI based classification. I strongly believe that MRI is required.

Ultrasound was performed by qualified board-certified musculoskeletal radiologist. We did not assess using MRI for our patients. However, as it has been previously shown by Colin et al MRI and US provide similar assessments of postoperative rotator cuff healing (Collin P, Yoshida M, Delarue A, Lucas C, Jossaume T, Lädermann A. Evaluating postoperative rotator cuff healing: Prospective comparison of MRI and ultrasound. Orthop Traumatol Surg Res 2015;101(6):S265-S8.).

Comment 2: Was the subscapolaris damaged before surgery in all cases?Or was it cut by the procedure itself?Also, the type of lesion might be different over the series.

The tenotomy was done on intact subscapularis tendon only which were then reduced to anatomic position prior to repair.

Tables are very disorganized and confusing.

Thank you for your comment. We decided that the demographic data and radiographic results required two tables instead of one. This addition of a new table to the existing ones necessitates renaming all the tables. Table 2, which we reorganized as per your suggestion, will now become Table 3; Table 3 will become Table 4, and Table 4 will become Table 5. We did modification in the manuscript"

Table 1 : Demographic data

Variable

Subcategory

Inlay RSA n (%)

Onlay RSA n (%)

Total n (%)

P-value

Sex

Male

35 (39)

37 (38)

72 (38)

P= 0.82

Female

54 (61)

61 (62)

115 (62)

Age

< 60

3 (17)

15 (83)

18 (10)

P= 0.04

60-69

27 (48)

29 (52)

56 (30)

70-79

43 (51)

42 (49)

85 (45)

≥ 80

16 (57)

12 (43)

28 (15)

Diagnosis

OA

24 (31)

54 (69)

78 (42)

P= 0.06

CTA

53 (78)

15 (22)

68 (36)

MCT

12 (29)

29 (71)

41 (22)

Legend: OA= Osteoarthritis, CTA= Cuff Tear Arthropathy, MCT= Massive Cuff Tear, RSA= Reverse Shoulder Arthroplasty

Table 2 : Radiographic results

Criteria

Subcategory

Healed n (%)

Non Healed n (%)

Total n (%)

P-value

RSA Design

Inlay

50 (56)

39 (44)

89 (100)

P= 0.013

Onlay

72 (73)

26 (27)

98 (100)

SSC Type

Type I

19 (15)*

0 (0)

19 (10)***

Type II

91 (75)*

0 (0)

91 (49)***

Type III

12 (10)*

0 (0)

12 (6)***

Type IV

0 (0)

12 (18)**

12 (6)***

Type V

0 (0)

53 (82)**

53 (29)***

Legend: RSA = Reverse Shoulder Arthroplasty, SSC= Subscapularis

*Purcentage of healed

**Purcentage of non healed

***Purcentage of RSA design

Table 3 : Healing of subscapularis tendon identified using US according to age group and

preoperative diagnosis

Criteria

Subcategory

Healed n (%)

Non Healed n (%)

Total n (%)

P-value

Age

< 60

14 (78)

4 (22)

18 (100)

P=0.190

60-69

38 (68)

18 (32)

56 (100)

70-79

49 (58)

36 (42)

85 (100)

≥ 80

21 (75)

7 (25)

28 (100)

Diagnosis

OA

58 (74)

20 (26)

78 (100)

P=0.059

CTA

42 (62)

26 (38)

68 (100)

MCT

22 (54)

19 (46)

41 (100)

Legend: OA; Osteoarthritis, CTA; Cuff Tear Arthropathy, MCT, Massive Cuff Tear.

Comment 3:Please provide graphic representation of inlay and outlay designs.

Thank you for your suggestions. We added graphic representations of inlay and onlay designs

Figure 1 : Onlay design

Figure 2: Inlay design

(Please see the attachment.)

Comment 4: Conclusions are not supported by results. These, are only hiopthesis of the Authors.

We showed that the healing rate of the repaired subscapularis tendon was 73% in the onlay group which was statistically significant compared to the healing rate in the inlay group (56%). With that in mind we wrote in the conclusion: The onlay systems may enhance subscapularis tendon healing compare to inlay ones, possibly due to preserved intramedullary vascularity and the near-normal tendon excursion that can be achieved by the onlay system.

Reviewer 2 Report

Comments and Suggestions for Authors

The theme of this retrospective study, apart from being ‘interesting’, presents with a clinical applicability solution for the surgeon, the physical therapist and the patient as well. Preserving the subscapularis integrity and function post-surgery adds stability and extra mobility towards internal rotation to the joint the implant was provided.

The design of the study is fine, and the main outcome seems to be the healing response of the subscapularis. It would have added to this study to know the range of motion as well as the functional (even with a self-report scale) improvement between the two groups of surgical management.

The authors have taken onto account several confounding factors that could have affected / biased between group differences, such as age, type of diagnosis and gender. Vascularity and biomechanical changes are proposed as an explanation of the results.

In general, the study is well-written and the authors are familiar with the intricacies of this type of surgery, as well as of the results’ explanation and the clinical significance of the findings. The statistical analysis is correct and the Discussion part is well-organised, presenting possible explanations of the findings, the factors that the study controlled for as well as its limitations.

Specific comments:

Line 114: Add the full word (ultrasound) before its abbreviation (US).

Please provide the explanation in full of all abbreviations that are within table 4

There is no way to verify whether the result is due to the physical therapy (line 233) received. Have the authors controlled in any way for this parameter that might have affected the results?

What do you mean by ‘the patient well-being (lines 232-3), and in what way could it have affected the results of the study?

The study's conclusion suggests that the patients may have increased internal rotation of the shoulder, despite not having it measured. Please change this.

Author Response

Thank you for your comments. Please see our answers below.

Comment 1: The design of the study is fine, and the main outcome seems to be the healing response of the subscapularis. It would have added to this study to know the range of motion as well as the functional (even with a self-report scale) improvement between the two groups of surgical management.

We understand your comment. In this study, we did not include the range of motion post RSA as we were solely looking at the healing rate of subscapularis tendon in 2 different RSA systems. In the next study, we will assess these parameters.

Comment 2: Line 114: Add the full word (ultrasound) before its abbreviation (US).

Done. Line 121: Ultrasound (US) Evaluation

Comment 3: Please provide the explanation in full of all abbreviations that are within table 4

Done. For information as we added the new Table as suggested by reviewer#1, Table 4 became Table 5.

Table 5 : Results binomial Logistic regression

Legend: SE= Standard error, Z= Z value ((or Z statistic), which measures the statistical significance of each predictor in the model) , p= P value, OA= Osteoarthritis, CTA= Cuff Tear Arthropathy, MCT=Massive Cuff Tear, F= Female, M= Male. Design:  1 (for onlay) and 0 for inlay. Side :1 for right and 0 for left

Comment 4: There is no way to verify whether the result is due to the physical therapy (line 233) received. Have the authors controlled in any way for this parameter that might have affected the results?

All patients underwent the same postoperative rehabilitation as described in the manuscript. All subscapularis tendon that is intact and healthy is repaired using the same method by a single surgeon, there are no other parameters that we assessed.

Comment 5: What do you mean by ‘the patient well-being (lines 232-3), and in what way could it have affected the results of the study?

 Thank you for your comment. You are right, this information is irrelevant. We Deleted it.

This is the new sentence, line 244-246: “There are also other factors that could be influencing the healing of the subscapularis tendon such as the bone and tendon quality”

Comment 6: The study's conclusion suggests that the patients may have increased internal rotation of the shoulder, despite not having it measured. Please change this

Thank your for your comment. You are right, we delete it.

Reviewer 3 Report

Comments and Suggestions for Authors

Dear Authors!

Thank you for the opportunity to review your manuscript.

The shoulder arthroplasty is one of the frequent orthopedic surgeries in elder people and finding the predictors of the outcomes is necessary.

The Authors provided clear goals, made the selection of the patients, and provided a flow-chart of the study

The methods are clear and reproducible.

The results confirmed by the tables and only minor changes are needed

The discussion contains the relevant and contemporary literature and the Authors compared their results with previously published literature data

The conclusion supports the study results

Manuscript has limitations section,where the Authors disclose the weak parts or the study

During the review, I have several comments

1) table 1 please provide the data in n (%) format.

2) Table 2 1st line "healed, n (%) and unhealed, n (%) and delete the "%" from the brackets. p-value - 3 digits in the decimals is enough

3) Table 4: first small table is not necessary, the data about r2 you can provide in the text or add to the main table 4.

Author Response

Reviewer #3:

Thank you for your comments. Please see our answers below.

Comment 1: table 1 please provide the data in n (%) format.

Thank you for your comment. We provided the data n (%) format and as suggested by reviwer#1 we reorganized this table in 2 Tables (Table and Table 2).

Table 1 : Demographic data

Variable

Subcategory

Inlay RSA n (%)

Onlay RSA n (%)

Total n (%)

P-value

Sex

Male

35 (39)

37 (38)

72 (38)

P= 0.82

Female

54 (61)

61 (62)

115 (62)

Age

< 60

3 (17)

15 (83)

18 (10)

P= 0.04

60-69

27 (48)

29 (52)

56 (30)

70-79

43 (51)

42 (49)

85 (45)

≥ 80

16 (57)

12 (43)

28 (15)

Diagnosis

OA

24 (31)

54 (69)

78 (42)

P= 0.06

CTA

53 (78)

15 (22)

68 (36)

MCT

12 (29)

29 (71)

41 (22)

Legend: OA= Osteoarthritis, CTA= Cuff Tear Arthropathy, MCT= Massive Cuff Tear, RSA= Reverse Shoulder Arthroplasty

Table 2 : Radiographic results

Criteria

Subcategory

Healed n (%)

Non Healed n (%)

Total n (%)

P-value

RSA Design

Inlay

50 (56)

39 (44)

89 (100)

P= 0.013

Onlay

72 (73)

26 (27)

98 (100)

SSC Type

Type I

19 (15)*

0 (0)

19 (10)***

Type II

91 (75)*

0 (0)

91 (49)***

Type III

12 (10)*

0 (0)

12 (6)***

Type IV

0 (0)

12 (18)**

12 (6)***

Type V

0 (0)

53 (82)**

53 (29)***

Legend: RSA = Reverse Shoulder Arthroplasty, SSC= Subscapularis

*Purcentage of healed

**Purcentage of non healed

***Purcentage of RSA design

Comment 2: Table 2 1st line "healed, n (%) and unhealed, n (%) and delete the "%" from the brackets. p-value - 3 digits in the decimals is enough

Thank you for your comment. We added a new Table, that is why this table you are talking about became Table 3.

We did modifications you suggested and we reorganized it as suggested by reviewer#1

Table 3 : Healing of subscapularis tendon identified using US according to age group and

preoperative diagnosis

Criteria

Subcategory

Healed n (%)

Non Healed n (%)

Total n (%)

P-value

Age

< 60

14 (78)

4 (22)

18 (100)

P=0.190

60-69

38 (68)

18 (32)

56 (100)

70-79

49 (58)

36 (42)

85 (100)

≥ 80

21 (75)

7 (25)

28 (100)

Diagnosis

OA

58 (74)

20 (26)

78 (100)

P=0.059

CTA

42 (62)

26 (38)

68 (100)

MCT

22 (54)

19 (46)

41 (100)

Legend: OA; Osteoarthritis, CTA; Cuff Tear Arthropathy, MCT, Massive Cuff Tear.

Comment 3: Table 4: first small table is not necessary, the data about r2 you can provide in the text or add to the main table 4.

As you suggested we deleted the first small table in this table. For information as suggested by reviewr#1 we added a new table that is why this table 4 you are talking about became Table 5

Table 5 : Results binomial Logistic regression

Legend: SE= Standard error, Z= Z value ((or Z statistic), which measures the statistical significance of each predictor in the model) , p= P value, OA= Osteoarthritis, CTA= Cuff Tear Arthropathy, MCT=Massive Cuff Tear, F= Female, M= Male. Design:  1 (for onlay) and 0 for inlay. Side :1 for right and 0 for left

Reviewer 4 Report

Comments and Suggestions for Authors

This is a well-structured article. The study is retrospective and it is not clear if the 2 groups were matched for characteristics such as comorbidities, occupation and recreational activities. The aim of this study is to saw if the type of the RSA design can influence the integrity of the subscapularis repair and the message in the conclusion is quite clear, favouring the on-lay design. The authors make good use of the tables, avoiding long and complex paragraphs, giving the necessary information in a clear way.

In order to decide if the hypothesis is relevant for the general orthopaedic practice, what needs to be clarified is whether the integrity of the subscapularis tendon affects the clinical outcome and whether this is measurable. It appears that integrity of subscapularis tendon in the RSA, improves the range of internal rotation, but the majority of the studies cannot demonstrate significant difference in functional or pain scores. Even postoperative complications such as dislocations, have not been strongly correlated to the repair status of the subscapularis. 

In the introduction paragraph (line 50-56) there is room for more literature discussion on the pros and cons of subscapularis integrity/repair in the RSA. 

It is known that some studies have demonstrated an antagonist action of the repaired subscapularis against the deltoid.  In lines 52-53, the short statement: ‘’ but it is also reported that it could decrease shoulder strength and function [8-11]’’ somehow conforms this.  The authors need to make an effort to analyse the literature and explain why they preferer repairing the tendon based on published outcomes. This way they will add value to their hypothesis and to their clear conclusion.

Another point that could help in this direction is the specific characteristics of the two groups, such as comorbidities and habits (diabetes, smoking etc.)  so, we could take more factors out of the equation of healing.

Line 89 -100: a drawing of the repair technique would be helpful. There is the appropriate reference (self-citation) in the text.

Finally, a question that comes in readers mind reading this article is whether a recommendation can be made regarding the necessity of subscapularis repair when we use an inlay implant. There is evidence that subscapularis repair works better in more varus RSA designs. Is it something that we could avoid with inlay implants? 

Thank you.

Author Response

Thank you for your comments. Please see our answers below.

Comment 1 :In order to decide if the hypothesis is relevant for the general orthopaedic practice, what needs to be clarified is whether the integrity of the subscapularis tendon affects the clinical outcome and whether this is measurable. It appears that integrity of subscapularis tendon in the RSA, improves the range of internal rotation, but the majority of the studies cannot demonstrate significant difference in functional or pain scores. Even postoperative complications such as dislocations, have not been strongly correlated to the repair status of the subscapularis.

The integrity of subscapularis tendon does improve the internal rotation and the strength and improve shoulder stability post RSA as quoted from reference 6 & 7.

Comment 2: It is known that some studies have demonstrated an antagonist action of the repaired subscapularis against the deltoid.  In lines 52-53, the short statement: ‘’ but it is also reported that it could decrease shoulder strength and function [8-11]’’ somehow conforms this.  The authors need to make an effort to analyse the literature and explain why they preferer repairing the tendon based on published outcomes. This way they will add value to their hypothesis and to their clear conclusion.

Thank you for your comment. Revised the whole sentence. Line 55-58 : Recent literature shows that repairing the subscapularis tendon in RSA can improve active internal rotation [6] especially in restoring the strength and enhance shoulder stability[7] but there are also reports showing that shoulder strength and function are not significantly impacted by this repair. [8-11]

Comment 3 :Another point that could help in this direction is the specific characteristics of the two groups, such as comorbidities and habits (diabetes, smoking etc.)  so, we could take more factors out of the equation of healing.

We did not include those parameters in our study this time. The next study might look into the factors that could affect healing.

Comment 4: Line 89 -100: a drawing of the repair technique would be helpful. There is the appropriate reference (self-citation) in the text.

Thank you for your comment. You are absolutely right; we have added drawings. For your information, at the request of the reviewer#1, we had already included two figures illustrating the prosthesis design. Therefore, with these new drawings added following your helpful suggestions, we had to rename the other figures. Figure 1 has now become Figure 8, and Figure 2 has now become Figure 9. We did modifications in the manuscript.

Line 139: Figure 8. Transosseous repaired of the subscapularis tendon in RSA.

Line 173: (Figure 9).

Figure 3 (A) A suture tape prefashioned with a half racking suture on the end is passed from lateral to medial through the inferior 2 holes, and (B) a separate suture is passed through the superior hole.

Figure 4 The stem is placed so that the sutures pass around the prosthesis. (A) The sutures are passed through the subscapularis tendon, and (B) the wedged ends are cut to provide access to 4 free limbs.

Figure 5 (A) One suture limb from each pair is selected and (B) passed through the prefashioned half racking suture.

Figure 6 The suture limbs passed through the half racking suture are tensioned. Tensioning was done under visual inspection.

Figure 7 Final repair.

(Please see the attachment.)

Comment 5: Finally, a question that comes in readers mind reading this article is whether a recommendation can be made regarding the necessity of subscapularis repair when we use an inlay implant. There is evidence that subscapularis repair works better in more varus RSA designs. Is it something that we could avoid with inlay implants?

The conclusion was made solely based on a theory and a proposed explanation. There is no biomechanical evidence yet. But we have modifcated our conclusion.

Line 387-389: Opting for onlay designs during RSA could minimize bone cuts and enhance subscapularis healing. Further research should explore how vascular and biomechanical factors interact to influence tendon healing

Round 2

Reviewer 1 Report

Comments and Suggestions for Authors

Despite the Authors' efforts, they were not able to address appropriately to most of my previous concerns.

The study still presents major biases.

Comments on the Quality of English Language

Still many grammar and syntax errors.